# Intraindividual variation of dose parameters in oncologic CT imaging

**Isabel Lange**[1]*, **Babak Alikhani**[2], **Frank Wacker**[1], **Hans-Juergen Raatschen**[1]

1 Department of Diagnostic and Interventional Radiology, Hannover Medical School, Hannover, Germany,
2 Center for Radiology and Nuclear Medicine, Diakovere Henriettenstift, Hannover, Germany

* isabel.lange@med.uni-jena.de

**Data Availability Statement:** All relevant data are within the manuscript.

**Funding:** The authors received no specific funding for this work.

## Abstract

The objective of this study is to identify essential aspects influencing radiation dose in computed tomography [CT] of the chest, abdomen and pelvis by intraindividual comparison of imaging parameters and patient related factors. All patients receiving at least two consecutive CT examinations for tumor staging or follow-up within a period of 22 months were included in this retrospective study. Different CT dose estimates (computed tomography dose index [CTDI$_{vol}$], dose length product [DLP], size-specific dose estimate [SSDE]) were correlated with patient's body mass index [BMI], scan length and technical parameters (tube current, tube voltage, pitch, noise level, level of iterative reconstruction). Repeated-measures-analysis was initiated with focus on response variables (CTDI$_{vol}$, DLP, SSDE) and possible factors (age, BMI, noise, scan length, peak kilovoltage [kVp], tube current, pitch, adaptive statistical iterative reconstruction [ASIR]). A univariate-linear-mixed-model with repeated-measures-analysis followed by Bonferroni adjustments was used to find associations between CT imaging parameters, BMI and dose estimates followed by a subsequent multivariate-mixed-model with repeated-measures-analysis with Bonferroni adjustments for significant parameters. A p-value <0.05 was considered statistically significant. We found all dose estimates in all imaging regions were substantially affected by tube current. The iterative reconstruction significantly influenced all dose estimates in the thoracoabdominopelvic scans as well as DLP and SSDE in chest-CT. Pitch factor affected all dose parameters in the thoracoabdominopelvic CT group. These results provide further evidence that tube current has a pivotal role and potential in radiation dose management. The use of iterative reconstruction algorithms can substantially decrease radiation dose especially in thoracoabdominopelvic and chest-CT-scans. Pitch factor should be kept at a level of ≥1.0 in order to reduce radiation dose.

## Introduction

Computed tomography (CT) is the main diagnostic modality in oncologic imaging and is widely used for the detection and follow-up of tumors. Although only 9% of all radiological examinations are CT scans, they contribute to up to 65% of the medically induced radiation exposure [1]. Consequently, the continuous reduction, optimization as well as inter- and

**Competing interests:** The authors have declared that no competing interests exist.

intraindividual consistency of dose are fundamental goals in radiology. An important aspect of dose optimization comes with the necessity for consistent image quality throughout multiple CT scans of the same patient and pathology, to reliably evaluate tumor response to treatment at exposure levels as low as reasonably achievable (ALARA principle).

In the recent years, several dose reduction techniques have been developed and are widely established: The automatic tube current modulation (ATCM) adjusts the tube current to the X-ray attenuation of the patient and represents one option to reduce radiation dose for patients while maintaining diagnostic image quality [2,3]. The tube voltage modulation presents a different option for potentially reducing the dose as well as keeping a satisfactory image quality [4,5]. For image reconstruction, the standard filtered back projection (FBP) methods are increasingly replaced by iterative reconstruction algorithms since it has been demonstrated repeatedly that they are able to reduce radiation dose while maintaining high image quality [6–8].

One of the most surprising observations after establishing a dose management software in our institution was a substantial intraindividual variation of certain dose estimates in our cohort of oncologic patients (Table 1). This was even more surprising, since all technologists adhered to standardized protocols defined in standard operation procedures (SOP) and saved at the CT machines. While the most obvious cause for dose variation, namely a significant change in body mass index (BMI) could be excluded, the relevance of other factors remained unclear at first sight. Thus, we intended to identify major parameters that account for the substantial intraindividual divergence in radiation exposure. We found the group of tumor patients particularly well suited for this goal, since this cohort typically receives several consecutive CT scans with comparable imaging protocols and settings within a short period of time.

## Material and methods

### 2.1 Study design

For this single center study, informed written consent was obtained for every single CT examination and scientific use of the data from all patients. Approval was waived by the institutional review board at Hannover Medical School due to the retrospective design of the study.

A dose management software (DoseWatch, GE Healthcare, USA) was used to identify studies suitable for intra- and interindividual comparison, to gather CT imaging parameters (scan length, peak kilovoltage (kVp), milliamperes (mAs), pitch factor, noise index, iterative reconstruction level), dose estimates (dose length product (DLP), computed tomography dose index ($CTDI_{vol}$) and size specific dose estimate (SSDE)) as well as patient's age, weight and height automatically from a dedicated data base.

At first, all patients receiving at least two comparable CT scans for tumor imaging during this period of time were identified. CT scans were considered comparable, when patients were imaged with a single-phase contrast-enhanced CT scan. CT scans with more than one single phase (e.g. biphasic liver, unenhanced/contrast-enhanced) were excluded from further analysis. Finally, we identified three different patient groups receiving CT imaging of the chest (group 1), the abdomen and pelvis (group 2) or chest, abdomen and pelvis (group 3) for tumor

**Table 1. Intraindividual DLP deviation in study population based on two consecutive staging CT examinations of the same body region.**

|  | Δ DLP | % |
|---|---|---|
| *Chest* | 40.78 ± 280.60 mGycm | (1.14 ± 64.61%) |
| *Abdomen/Pelvis* | 16.03 ± 696.93 mGycm | (23.92 ± 127.53%) |
| *Chest/Abdomen/Pelvis* | 15.85 ± 367.48 mGycm | (3.60 ± 41.76%) |

staging. All of our scans were performed at the two CT scanners used for tumor imaging at our institution (Lightspeed 16 and Lightspeed VCT, GE Healthcare, USA). Based on the appointment and the clinical question, patients were assigned to one of the two CT scanners by the medical assistants at the front desk.

## 2.2 Patient population

Our database consists of all CT scans performed at the institution between March 2013 and December 2014. For an intraindividual analysis of our study, we required patients who received at least 2 CT scans or more during this period. We identified 242 patients in the chest group, 136 patients in the abdomen/pelvis group and 581 patients in the chest/abdomen/pelvis group that received at least two consecutive CT scans of the same body region (Fig 1). We only compared results from the first two CT scans since the number of patients receiving more

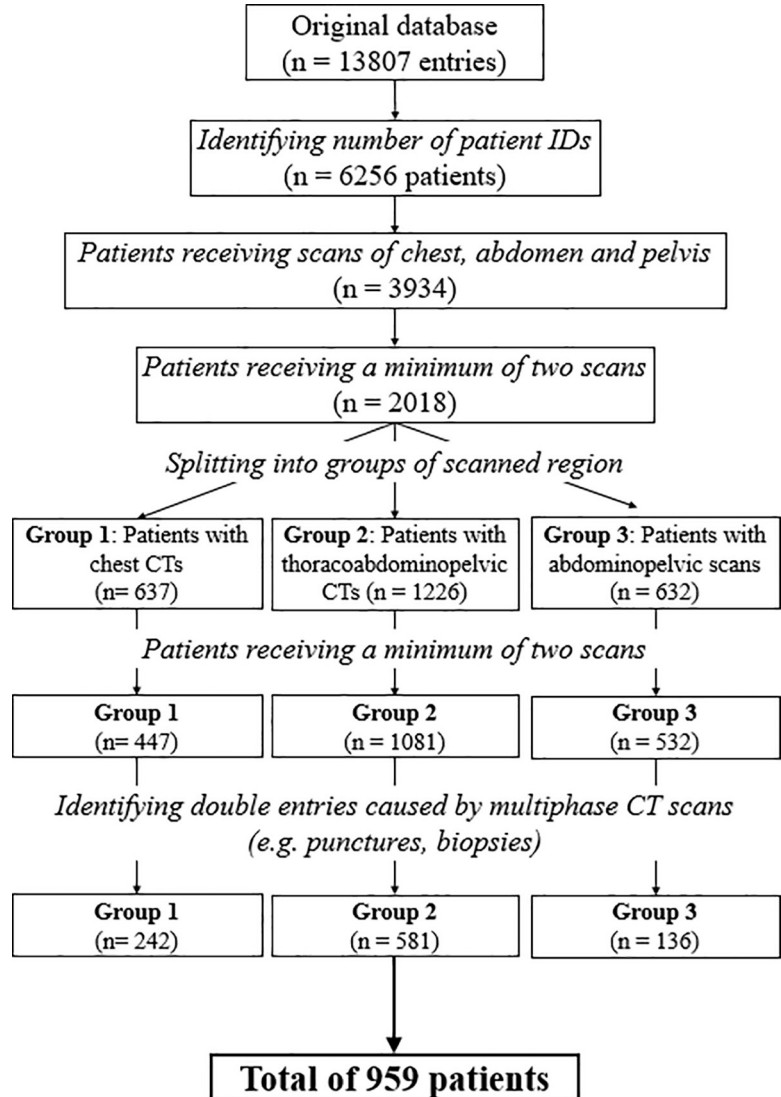

**Fig 1. Flow diagram of the process filtering the included study population.** Original database included all scans from March 2013 to December 2014 and was reduced and grouped to CTs of the chest, abdomen and pelvis.

than two examinations was negligible in all three groups. Thus, the total number of examinations analyzed in this study was n = 1918.

The most frequent indications for CT tumor imaging were: lymphoma, pulmonary metastases, and different tumors including lung, hepatocellular, colon, esophageal, pancreatic and prostate cancer as well as malignant melanoma and others.

Data was acquired with two CT scanners (LightSpeed 16; General Electric Healthcare, USA and LightSpeed VCT; General Electric Healthcare, USA) used at our institution for oncologic imaging. All scans were performed with automatic exposure control techniques as both GE scanners in use have techniques for the protocols involved that are based on tube current modulation through the noise index [9]. All scans of the combined chest and abdomen group used automated exposure control with a noise index between 15.5 and 22; this protocol was performed in a single phase from cranial (starting at the level of the upper thoracic aperture) to caudal (ending at the level of the lower pubic branch).

During the time period of the CT examinations, one of our CT scanners was upgraded with an iterative reconstruction algorithm (adaptive statistical iterative reconstruction, ASIR). However, no other protocol changes have been made during this time period: all other imaging parameters (e.g. tube voltage, noise index, pitch, tube current modulation) were kept identical and technologists adhered to protocol parameters defined in standard operation procedures (SOP).

## 2.3 Radiation dose parameters

The dose estimates used for comparison were the mean computed tomography dose index ($CTDI_{vol}$), the dose length product (DLP) and the size-specific dose estimate (SSDE). These measured or calculated variables were correlated with the scan length and body mass index (BMI) of the patients along with the tube current, tube voltage, pitch, noise level and the level of iterative reconstruction.

BMI was calculated from the patient's body weight and height and was only used as a general factor to measure adiposity and how much the individual's body weight differs from the normal weight for a person of the same height [10].

Length of the CT scan was selected by the technologist from the scout image depending on the chosen protocol and region of interest.

## 2.4 Statistical analysis

The data was analyzed using SPSS Version 24 (IBM). Mean, standard deviation (mean ± SD) median, 25% and 75% quantile, as well as maxima and minima were calculated for age of the patient at the time of the scan, BMI, DLP, scan length, kVp, tube current, $CTDI_{vol}$, pitch, noise, SSDE and adaptive statistical iterative reconstruction (ASIR).

For the intraindividual analysis, we calculated the difference between the second and the first CT scan for each group, each response and each continuous variable. For categorical variables, we calculated the relative difference between the second and first scan by dividing the second measurement by the first one. Consecutively, we applied a univariate linear mixed noise model with repeated measures analysis to the values followed by Bonferroni adjustments of the confidence intervals to find associations between each individual averaged continuous variable and each categorical variable for each of the radiation dose variables. In the case of significant results in this univariate analysis, we proceeded with a subsequent multivariate linear regression model with repeated measures analysis including all variables that had shown statistically significant results in the univariate analysis followed by Bonferroni adjustments. The results are estimates with 95% confidence intervals and p-values for all the univariate and

multivariate analysis. These estimates imply either a decrease or increase of the analyzed dose parameter ($CTDI_{vol}$, DLP or SSDE) depending on the algebraic sign, e.g. a pitch increase by 1 unit leads to a SSDE decline of 3.15 mGy.cm (Table 2). A p-value of less than 0.05 was considered statistically significant.

## Results

### 3.1 Imaging parameters

Tube voltage (kVp), pitch factor, noise index and iterative reconstruction level (ASIR) were mostly constant throughout all three organ groups, which is why they are described here for all three organ groups together:

The tube voltage had three different values: 100, 120 or 140 kVp, where 120 kVp was mostly in use.

The pitch factor ranged from 1.75 to 1.38, 0.98, 0.63 (only abdomen) and 0.52.

Noise index was mostly fixed at 22 for chest CT, but a small number of was performed with a noise index of 13, 40 or 44. For the abdominopelvic CT group three different noise indices were used: 18, 40 or 44. In the thoracoabdominopelvic group the noise index was set to 18 or 15.5.

Concerning the use of the two different machines, we found that overall 63% of all 959 patients underwent CT imaging at the same CT scanner for both points in time. If CT scanners changed between both points in time, the most common variation was the first scan being done with the LightSpeed 16 and the second with the VCT.

For the 242 patients receiving chest CT scans, 112 received both scans with the LightSpeed VCT, 43 with exclusively the LightSpeed 16 and 87 received the first with the LightSpeed 16 and the second CT at the LightSpeed VCT.

The abdomen group consisted of 52 patients receiving both scans with the LightSpeed VCT, 51 with the LightSpeed 16 and 33 patients received scans from different scanners.

581 patients who received a combined abdomen and thorax scan showed the following: 242 patients were scanned with the LightSpeed VCT, 103 with the Lightspeed 16, 83 with LightSpeed VCT first and 16 second, 153 LightSpeed 16 first and VCT second.

**Table 2. Results of intraindividual analysis of chest CTs for the dose estimate parameter CTDIvol, DLP and SSDE.**

| Characteristic | Univariate CTDIvol | | Multivariate CTDIvol | | Univariate DLP | | Multivariate DLP | | Univariate SSDE | | Multivariate SSDE | |
|---|---|---|---|---|---|---|---|---|---|---|---|---|
| | Estimate (95% CI) | p-value | Estimate (95% CI) | p-value | Estimate (95% CI) | p-value | Estimate (95% CI) | p-value | Estimate (95% CI) | p-value | Estimate (95% CI) | p-value |
| *BMI difference* | .132 (-.024; .288) | .10 | NA | | 3.52 (-2.05; 9.08) | .21 | NA | | .128 (-.068; .325) | .20 | NA | |
| *scan length difference* | -.011 (-.023; .002) | .10 | NA | | .258 (-.200; .716) | .27 | NA | | -.006 (-.022; .010) | .46 | NA | |
| *tube current difference* | .019 (.014; .025) | < .05 | .039 (.034; .043) | < .05 | .649 (.451; .847) | < .05 | 1.124 (.883; 1.364) | < .05 | .018 (.011; .025) | < .05 | .043 (.036; .050) | < .05 |
| *kVp difference* | 23.03 (15.74; 30.32) | < .05 | 48.88 (43.24; 54.52) | < .05 | 788.63 (527.98; 1049.28) | < .05 | 1534.69 (1232.07; 1837.32) | < .05 | 28.40 (19.21; 37.59) | < .05 | 42.95 (34.06; 51.85) | < .05 |
| *pitch difference* | -2.10 (-4.60; .40) | .10 | NA | | -88.15 (-176.78; .47) | .051 | NA | | -3.15 (-6.29; -.02) | < .05 | -9.04 (-11.87; -6.22) | < .05 |
| *noise difference* | 1.59 (-2.77; 5.96) | .47 | NA | | 40.11 (-115.13; 195.34) | .61 | NA | | 2.54 (-2.95; 8.02) | .36 | NA | |
| *iterative reconstruction difference* | -1.88 (-3.76; .002) | .050 | NA | | -69.85 (-135.14; -4.56) | < .05 | -56.73 (-104.65; -8.82) | < .05 | -2.21 (-4.36; -.07) | < .05 | -1.46 (-2.81; -.11) | < .05 |

The use of iterative reconstruction algorithms (ASIR) depended on the CT scanner that was in use, since only one of the machines was equipped with this technique. If iterative reconstruction was applied, it was set at the level of either 30% or 50%.

## 3.2 Intraindividual analysis

**3.2.1 Chest CT.**   In the patient population for the chest CT, we found an age range from 16 to 81 years with an average of 54.4 ± 16 years. The average BMI was 25.24 ± 6.24 kg/m$^2$. Scan length was found with a value of 350.49 ± 38.24 mm. The average tube current was 191.06 ± 82.23 mA. Dose parameters yielded the following results: CTDI$_{vol}$ 8.92 ± 3.70 mGy; DLP 318.85 ± 135.02mGy·cm and SSDE 11.22 ± 3.84 mGy.

In the univariate analysis, tube current and tube voltage yielded significant CTDI$_{vol}$ differences. Tube current, tube voltage as well as the level of iterative reconstruction yielded significant differences in DLP values. SSDE was influenced significantly by tube current, tube voltage, pitch factor, noise index and the level of iterative reconstruction.

In the multivariate analysis, tube current and tube voltage contributed significantly for CTDI$_{vol}$ differences. DLP was significantly influenced by tube current, tube voltage and the level of iterative reconstruction. SSDE differences were significantly related to tube current, tube voltage, pitch factor, noise index and the level of iterative reconstruction (Table 2)

**3.2.2 Abdomen/Pelvis CT.**   The patients of the abdomen/pelvis group had an age span from 16 to 87 years with an average of 59 ± 16.83 years. The average BMI was 25.65 ± 5.28 kg/m$^2$. Scan length was 475.69 ± 101.37 mm. The tube current had an average of 264.69 ± 85.55 mA.

The dose parameters were as follows: CTDI$_{vol}$ 12.55 ± 4.22 mGy; DLP 607.83 ± 256.33mGy·cm and SSDE 15.75 ± 3.82mGy. In the univariate analysis, CTDI$_{vol}$, was significantly influenced by tube current, tube voltage, pitch factor and noise index. DLP was significantly correlated with scan length, tube current, pitch factor and noise index. SSDE was found to be depending on scan length, tube current, tube voltage and pitch factor.

In the multivariate analysis, changes in tube current, tube voltage and pitch contributed significant for CTDI$_{vol}$ changes. DLP was significantly influenced by scan length and tube current, and SSDE differences were significantly related to changes in tube current, tube voltage and pitch factor (Table 3).

**3.2.3 Chest/Abdomen/Pelvis CT.**   Patient's age ranged from 18 to 91 with an average of 62 ± 12.75 years. The average BMI was 26.11 ± 5.16 kg/m$^2$. Mean scan length was 676.75 ± 62.72 mm. The tube current averaged at 225.51 ± 86.45mA. The dose parameters yielded the following results: CTDI$_{vol}$ 12.22 ± 4.06mGy; DLP 841.45 ± 301.23 mGy·cm and SSDE 15.07 ± 3.83mGy. In the univariate analysis, CTDI$_{vol}$ was significantly influenced by scan length, tube current, pitch and iterative reconstruction level. Tube current, pitch and levels of iterative reconstruction resulted in significant DLP differences. Changes in scan length, tube current, pitch and iterative reconstruction levels contributed significantly for SSDE differences. In the multivariate analysis, tube current, pitch and iterative reconstruction levels showed significant influence on CTDI$_{vol}$. Tube current, pitch and levels of iterative reconstruction resulted in significant differences for DLP. Tube current, pitch and levels of iterative reconstruction contributed to significant SSDE differences (Table 4).

## Discussion

Oncological patients often receive several consecutive CT scans for tumor staging. One of the first things we observed after establishing a software for systematic dose management was a substantial intraindividual variation in DLP of up to 23.9%. Since the optimization of imaging

**Table 3. Results of intraindividual analysis for CT scans of the abdomen and pelvis for the dose estimate parameter CTDIvol, DLP and SSDE.**

| Characteristics | Univariate CTDIvol | | Multivariate CTDIvol | | Univariate DLP | | Multivariate DLP | | Univariate SSDE | | Multivariate SSDE | |
|---|---|---|---|---|---|---|---|---|---|---|---|---|
| | Estimate (95% CI) | p-value | Estimate (95% CI) | p-value | Estimate (95% CI) | p-value | Estimate (95% CI) | p-value | Estimate (95% CI) | p-value | Estimate (95% CI) | p-value |
| *BMI difference* | -.036 (-.296; .224) | .79 | NA | | -4.57 (-19.48; 10.35) | .55 | NA | | -.136 (-.496; .224) | .45 | NA | |
| *scan length difference* | -.005 (-.011; .002) | .14 | NA | | 1.08 (.76; 1.41) | < .05 | 1.21 (.96; 1.45) | < .05 | -.009 (-.018; -.001) | < .05 | -.003 (-.008; .002) | .21 |
| *tube current difference* | .040 (.035; .045) | < .05 | .039 (.034; .044) | < .05 | 1.77 (1.38; 2.16) | < .05 | 1.82 (1.49; 2.14) | < .05 | .048 .041; .055) | < .05 | .045 (.038; .051) | < .05 |
| *kVp difference* | 30.13 (9.62; 50.64) | < .05 | 37.87 (26.81; 48.93) | < .05 | 1030.95 (-168.77; 2230.67) | .09 | NA | | 31.71 (5.84; 57.57) | < .05 | 42.23 (27.34; 57.11) | < .05 |
| *pitch difference* | 9.29 (5.02; 13.55) | < .05 | 2.85 (.31; 5.40) | < .05 | 329.26 (74.43; 594.10) | < .05 | 132.82 (-30.54; 296.18) | .11 | 14.11 (8.66; 19.55) | < .05 | 6.85 (3.21; 10.50) | < .05 |
| *noise difference* | -7.04 (-12.36; -1.72) | < .05 | .20 (-2.82; 3.22) | .90 | -599.44 (-886.35; -312.53) | < .05 | 63.19 (-144.79; 271.17) | .55 | -9.001 (-18.937; .935) | .08 | NA | |
| *iterative reconstruction difference* | .181 (-1.469; 1.831) | .83 | NA | | 35.43 (-59.03; 129.89) | .46 | NA | | .53 (-1.56; 2.61) | .62 | NA | |

parameters is a constant topic in computed tomography, especially with its constant increase in use, we intended to identify factors possibly influencing radiation dose in a large cohort of patients admitted for tumor staging CT. We chose to focus on the three most commonly used dose parameters CTDI$_{vol}$, DLP and SSDE in typical CT scan protocols used for tumor imaging (chest CT, abdominopelvic CT, thoracoabdominopelvic CT).

**Table 4. Results of intraindividual analysis for the CT scans of the thoracoabdominopelvic region for the dose estimate parameter CTDIvol, DLP und SSDE.**

| Characteristic | Univariate CTDIvol | | Multivariate CTDIvol | | Univariate DLP | | Multivariate DLP | | Univariate SSDE | | Multivariate SSDE | |
|---|---|---|---|---|---|---|---|---|---|---|---|---|
| | Estimate (95% CI) | p-value | Estimate (95% CI) | p-value | Estimate (95% CI) | p-value | Estimate (95% CI) | p-value | Estimate (95% CI) | p-value | Estimate (95% CI) | p-value |
| *BMI difference* | .035 (-.008; .079) | .11 | NA | | 2.853 (-.117; 5.824) | .06 | NA | | .039 (-.033; .111) | .29 | NA | |
| *scan length difference* | -.014 (-.019; -.009) | < .05 | -.002 (-.003; .000) | .09 | .159 (-.179; .498) | .36 | NA | | -.012 (-.020; -.004) | < .05 | -.003 (-.006; .001) | .12 |
| *tube current difference* | .031 (.029; .033) | < .05 | .058 (.055; .061) | < .05 | 2.012 (1.877; 2.147) | < .05 | 3.52 (3.22; 3.82) | < .05 | .035 (.031; .040) | < .05 | .063 (.058; .068) | < .05 |
| *kVp difference* | 1.58 (-13.95; 17.11) | .84 | NA | | 131.31 (-928.48; 1191.09) | .81 | NA | | -.478 (-26.000; 25.044) | .97 | NA | |
| *pitch difference* | 7.15 (6.01; 8.29) | < .05 | -16.25 (-17.74; -14.76) | < .05 | 451.13 (372.06; 530.19) | < .05 | -996.35 (-1161.20; -831.50) | < .05 | 8.34 (6.35; 10.32) | < .05 | -17.54 (-20.35; -14.73) | < .05 |
| *noise difference* | -.069 (-2.577; 2.439) | .96 | NA | | -45.81 (-216.94; 125.33) | .60 | NA | | -1.232 (-5.353; 2.889) | .56 | NA | |
| *iterative reconstruction difference* | -3.289 (-3.996; -2.581) | < .05 | -2.10 (-2.65; -1.55) | < .05 | -221.93 (-271.14; -172.71) | < .05 | -152.66 (-213.32; -91.99) | < .05 | -4.400 (-5.275; -3.526) | < .05 | -3.025 (-4.057; -1.992) | < .05 |

While some of the parameters analyzed in our study are patient related and thus unmodifiable at the time of imaging (e.g. age and BMI), others like tube current or the iterative reconstruction level are determined in the CT scan protocol. A further group of parameters (e.g. tube voltage and scan length) is in a certain range selected by the staff and thus more prone to individual variation. This last group of parameters seems to be highly important in regards to dose variation and could serve as a point of action for active dose management and optimization as well as for staff training.

The tube current plays an important role in maintaining image quality depending on the size of the patient and has been shown to be linearly related to radiation dose [11]. This relationship could be confirmed in our analysis, where we were able to show that tube current significantly influenced intraindividual dose estimates, namely $CTDI_{vol}$, DLP and SSDE in all three groups of patients and thus confirms its crucial role and importance for radiation dose management.

Furthermore, we found that every dose parameter and organ group in our study correlated significantly with patient's BMI. This correlates with the knowledge that overweight and obese patients absorb more radiation than patients with a normal BMI and thus require adjustments in the settings of the scanner to keep image quality adequate [10,12]. Our patients did not undergo substantial weight changes between their CT scans, thus the intraindividual analysis remained non-significant.

Another aspect that needs to be discussed is the iterative reconstruction technique. This algorithm has been shown to decrease patient's dose significantly in pediatric and adult patients undergoing chest CT or abdominopelvic CT [8,13,14]. According to these published studies, the adaptive statistical iterative reconstruction (ASIR) algorithm used at one of our two CT scanners significantly influenced $CTDI_{vol}$, DLP and SSDE estimates in the combined thoracoabdominopelvic CT as well as DLP and SSDE in chest CT with a substantial decrease in radiation dose in case of its use.

One aspect that has not been addressed is the time interval for response evaluation–modern oncological therapeutics require very short intervals (e.g. 4–6 weeks) between imaging for response evaluation and increase radiation dose by the factor 2 compared to conventional chemotherapy (normally 3 months). To reduce radiation dose, the number of image acquisitions during one examination has to be reduced to the necessary–not every tumor needs non-contrast or arterial phase imaging for response evaluation. On the other hand, the risk of radiation exposure has to be balanced to the survival rates of the tumor–while the 5-year survival rate of most metastasized cancers is below 10%, some tumors like lymphomas have much longer survival rates. This has to be considered when choosing time intervals for response evaluation as well as the number of imaging phases.

One limitation of this study is that not all patients were consistently scanned at the same CT at follow-up imaging, thus factors like differences in the x-ray tube or detector sensitivity have not been addressed. However, this difference is taken into account by our statistical analysis and by considering different noise, pitch and iterative reconstruction parameters.

Another limitation is that $CTDI_{vol}$ as a dose parameter does not regard the attenuation variations of a scan. Khatonabadi et al showed that using regional $CTDI_{vol}$ descriptors specify dose estimates more accurately than global $CTDI_{vol}$ values, in scans with tube current modulation [15]. A third limitation is the retrospective character of our study. However, this approach enabled us to identify main factors responsible for dose variation and helped to improve radiation dose consistency in our department. Furthermore, effects of patient positioning in the gantry (i.e. deviation from the isocenter) on radiation dose have not been evaluated, since this data was not available in our dose management software.

A tube voltage increase in consecutive intraindividual scans led to significant gain in DLP, $CTDI_{vol}$ and SSDE in chest CT, as well as to an increase in $CTDI_{vol}$ and SSDE in

abdominopelvic CT scans. This has to be considered when tube voltage needs to be adjusted in order to improve image quality in overweight patients [11]. On the other hand, lightweight patients could potentially benefit from a reduction of standard tube voltage settings from 120 kV to 100 kV or even 80 kV. This approach is further supported by Padole et al who showed that the image noise reduction abilities of iterative reconstruction techniques can be utilized to lower tube voltage in chest, abdomen and pediatric CT [16]. Thus, instead of adjusting tube voltage to higher levels in overweight, high BMI patients, iterative reconstruction techniques should be taken into account to improve image quality without increasing radiation dose.

The pitch factor is a further essential parameter for dose management. A pitch factor less than 1.0 leads to an overlap of x-ray beams, which results in a higher local and overall radiation dose. In our study, $CTDI_{vol}$, DLP and SSDE were significantly affected by the pitch factor in the thoracoabdominopelvic CT scans. As a result, pitch factor should be kept at a level of $\geq 1.0$ wherever possible.

## Conclusion

In conclusion, the dosage varied intraindividually up to 24% in CT examinations for staging. Dose management and consistency is a topic that involves staff operating the CT as well as the CT manufacturers. Awareness is a key to aim for a balance between radiation risk and accurate diagnostics. Staff performing CT studies can diminish and—in the best case—prevent unnecessary intraindividual dose variation when adhering to defined scan protocols, keeping tube current at the lowest possible level, using iterative reconstruction algorithms and avoiding pitch factors $< 1$.

## Author Contributions

**Conceptualization:** Hans-Juergen Raatschen.

**Data curation:** Isabel Lange.

**Formal analysis:** Hans-Juergen Raatschen.

**Investigation:** Isabel Lange.

**Methodology:** Babak Alikhani.

**Project administration:** Hans-Juergen Raatschen.

**Resources:** Isabel Lange.

**Supervision:** Frank Wacker.

**Writing – original draft:** Isabel Lange.

**Writing – review & editing:** Hans-Juergen Raatschen.

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
