## [Decision Letter · Decision Letter 0]

27 Nov 2020

PONE-D-20-27686

Intraindividual variation of dose parameters in oncologic CT imaging

PLOS ONE

Dear Dr. Lange,

Thank you for submitting your manuscript to PLOS ONE. After careful consideration, we feel that it has merit but does not fully meet PLOS ONE’s publication criteria as it currently stands. Therefore, we invite you to submit a revised version of the manuscript that addresses the points raised during the review process.

We look forward to receiving your revised manuscript.

Kind regards,

Rubens Chojniak, M.D., Ph.D.

Academic Editor

PLOS ONE

Journal Requirements:

Additional Editor Comments:

The authors studied factors that influence dosimetry parameters in computed tomography exams in patients who underwent at least 2 computed tomography exams. There were 242 patients. The results show that some known factors influence the dose received by the patient, such as iterative reconstruction, tube current and pitch. The authors results reinforce information useful in the control of radiation dose in diagnostic tests.

The article is well structured and written. I congratulate the authors.

Reviewers' comments:

Reviewer's Responses to Questions

**Comments to the Author**

1. Is the manuscript technically sound, and do the data support the conclusions?

Reviewer #1: Yes

2. Has the statistical analysis been performed appropriately and rigorously? 

Reviewer #1: Yes

3. Have the authors made all data underlying the findings in their manuscript fully available?

Reviewer #1: Yes

4. Is the manuscript presented in an intelligible fashion and written in standard English?

Reviewer #1: Yes

5. Review Comments to the Author

Reviewer #1: Dear authors,

This is an interesting article that addresses an important topic in clinical practice. The article is very well structured from the title to the conclusions, including the limitations of a retrospective cross-sectional cohort study. However, please consider reviewing the following aspects:

1- Regarding the study population, consider if it is not interesting to know how many patients were excluded from the analysis in the period evaluated. Consider using a flow diagram.

2- In the discussion, describe other limiting factors of the study, such as retrospective analysis and use of different tomographic equipment, since they are potential confounders and effect modifiers.

Congratulations on the job.

6. PLOS authors have the option to publish the peer review history of their article (what does this mean?). If published, this will include your full peer review and any attached files.

Reviewer #1: No

---

## [Author Response · Author response to Decision Letter 0]

26 Jan 2021

We thank you and the reviewers for a thorough reading and constructive criticism of our manuscript and for the opportunity to revise and resubmit. 

We are pleased to submit the improved research article titled “Intraindividual variation of dose parameters in oncologic CT imaging” for your consideration of publication in PLOS ONE.

On the following page, you will find our response to the editor and reviewer comments.

On behalf of my co-authors, I thank you for your consideration of this resubmission. We appreciate your time and look forward to your response.

Sincerely, 

Isabel Lange 

Editor Comments:

Reply of the authors: We have considered and applied all style and file naming requirements by PLOS ONE for this submission. 

2. PLOS requires an ORCID iD for the corresponding author in Editorial Manager on papers submitted after December 6th, 2016.

Reply of the authors: The corresponding author has created an ORCID iD: https://orcid.org/0000-0002-4069-6634

Reviewer Comments: 

1- Regarding the study population, consider if it is not interesting to know how many patients were excluded from the analysis in the period evaluated. Consider using a flow diagram.

Reply of the authors: We have created a flow diagram that presents additional information about the process of filtering the raw database to find the study population included in this study. The figure can be found under Material and Methods. 

2- In the discussion, describe other limiting factors of the study, such as retrospective analysis and use of different tomographic equipment, since they are potential confounders and effect modifiers.

Reply of the authors: This limitation has been added and is further elaborated on in our discussion.

---

## [Editor Report · Decision Letter 1]

8 Apr 2021

Intraindividual variation of dose parameters in oncologic CT imaging

PONE-D-20-27686R1

Dear Dr. Lange,

We’re pleased to inform you that your manuscript has been judged scientifically suitable for publication and will be formally accepted for publication once it meets all outstanding technical requirements.

Kind regards,

Rubens Chojniak, M.D., Ph.D.

Academic Editor

PLOS ONE

Additional Editor Comments (optional):

I congratulate the authors for the study submission and thank them for addressing the suggestions.
---

## [Editor Report · Acceptance letter]

15 Apr 2021

PONE-D-20-27686R1 

Intraindividual variation of dose parameters in oncologic CT imaging 

Dear Dr. Lange:

I'm pleased to inform you that your manuscript has been deemed suitable for publication in PLOS ONE. Congratulations! Your manuscript is now with our production department. 

Kind regards, 

on behalf of

Dr. Rubens Chojniak 

Academic Editor

PLOS ONE